# Comparison of West Nile Virus Disease in Humans and Horses: Exploiting Similarities for Enhancing Syndromic Surveillance

**DOI:** 10.3390/v15061230

**Published:** 2023-05-24

**Authors:** Erika R. Schwarz, Maureen T. Long

**Affiliations:** 1Montana Veterinary Diagnostic Laboratory, MT Department of Livestock, Bozeman, MT 59718, USA; erika.schwarz@mt.gov; 2Department of Comparative, Diagnostic, & Population Medicine, College of Veterinary Medicine, University of Florida, Gainesville, FL 32610, USA

**Keywords:** flavivirus, West Nile virus, human, equine, clinical symptoms, pathology, diagnosis, serology

## Abstract

West Nile virus (WNV) neuroinvasive disease threatens the health and well-being of horses and humans worldwide. Disease in horses and humans is remarkably similar. The occurrence of WNV disease in these mammalian hosts has geographic overlap with shared macroscale and microscale drivers of risk. Importantly, intrahost virus dynamics, the evolution of the antibody response, and clinicopathology are similar. The goal of this review is to provide a comparison of WNV infection in humans and horses and to identify similarities that can be exploited to enhance surveillance methods for the early detection of WNV neuroinvasive disease.

## 1. Introduction

Before its emergence in North America, West Nile virus (WNV) generally caused a febrile syndrome referred to as West Nile fever (WNF) with limited outbreaks of neuroinvasive disease in Africa and hotspots of activity in the Mediterranean and Eastern European countries [1,2,3,4]. Since the explosive emergence of WNV disease in North America, humans and horses are at risk for developing grave disease worldwide [5]. Infection in humans and horses, naturally outbred mammalian hosts, causes debilitating neurological disease and death with much commonality, but often, initial cases are missed due to lag time and gaps in reporting of environmental activity [6,7]. To enhance real-time detection of WNV activity in humans and horses, human and animal syndromic surveillance has been explored for its use in the prediction of WNV outbreaks [8,9,10,11,12,13]. Syndromic surveillance is a strategy for the identification of general health abnormalities not dependent on diagnostic testing to serve as an early warning system for disease threats [8,9,10,11,12,13]. This type of surveillance can be further optimized by a comprehensive understanding of the clinicopathological similarities of WNV in the two most affected mammalian species.

## 2. Virus Classification and Structure

West Nile virus is a mosquito-borne virus of the genus flavivirus, family Flaviviridae, first isolated in the West Nile district of Uganda in 1937 [14]. WNV is a spherical, enveloped, single-stranded, positive-sense RNA virus measuring 45–50 nm in diameter [15,16,17]. The 11 kb genome contains a single open reading frame (ORF) that is translated in its entirety into a polyprotein that is then cleaved by both cell and viral proteases into 11 viral proteins consisting of three structural proteins, including capsid (C), pre-membrane (prM)/membrane (M), envelope (E), and seven nonstructural (NS) proteins (NS1, NS2A, NS2B, NS3, NS4A, NS4B, and NS5) [17,18,19].

## 3. Epidemiology

Prior to the encroachment in North America, outbreaks of WNV neuroinvasive disease in humans or horses were becoming more frequent, as reported in North Africa, France, Romania, and Russia in the late 1990s [20,21,22,23]. With new geographic encroachments of the lineage 1 (L1) strain of WNV, high numbers of neuroinvasive infections, often with severe outcomes, were reported [5,24]. Seven lineages of WNV have been characterized, with L1a, L1b (Kunjin virus, KUN), and L2 documented in humans and horses [19,25,26,27,28,29,30]. While L2 was historically associated with WNF in humans before the 1990s and limited neurological disease in horses, a neuroinvasive strain of L2 emerged in Europe in the early 2000s, which was responsible for multiple outbreaks [29,31,32,33]. Japanese encephalitis virus (JEV) and Murray Valley encephalitis virus (MVEV) also cause disease in horses and humans [25,26]. WNV is now endemic on all continents except for Antarctica. Significant differences in nucleotide sequences between WNV isolates from different parts of the world have been documented along with antigenic variation among strains isolated within the same geographic region [6,34,35,36]. Continued expansion of multiple lineages and subtypes, with ongoing emergence and re-emergence, indicates that WNV disease will continue to evolve and threaten the health and well-being of both horses and humans worldwide [34,37].

### 3.1. Risk Factors for Human and Horse Exposure

#### 3.1.1. Macroscale Factors

Activity and identification of drivers of WNV transmission in human and horse outbreaks are variable and dependent upon the scale of the analysis [38]. Macroscale modeling of the abiotic and biotic risk factors for WNV transmission is generally predictive, overlapping as risk factors for infection in both humans and horses and predicated by the fact that these enhance exposure to competent vectors of WNV that feed on mammalian hosts in an environment with avian reservoirs [38,39]. Regardless of scale, climatological, hydrological, and altitudinal variables are the main abiotic factors, while avian and mosquito fauna, as well as vegetation, are common biotic factors [40,41,42,43,44,45,46,47,48,49,50,51]. In Europe, bird migration routes, maximum monthly temperature, and annual temperature, wet agricultural activities, presence of poultry and horses, presence of rivers, and lower altitudes have been identified as primary predictors for WNV activity, severity, and presence of human and/or equine cases [13,43,44,49,50,52,53,54,55]. Most of these same factors are identified as predictors of WNV activity in the Middle East and the Mediterranean coast of Africa [39,43,56,57]. In one study conducted in Africa, risk was modeled separately for enzootic and epizootic activity, a distinction not often considered in modeling [58]. Variables that increased enzootic activity included birds, temperature, proximity to RAMSAR-designated wetlands, flooded vegetation, and agriculture [58]. Predictors for epizootic activity were similar to enzootic activity; however, the presence of vectors, reservoir hosts, population levels, proximity to railways, and presence of irrigated crops were also predictors of outbreaks [58]. Mosquito and environmental data have been used for forecasting WNV activity in areas of high risk in the United States [41,59,60,61,62,63,64,65]. One of the largest and broadest scale US studies performed to date evaluated the risk for equine disease (as reported to state and federal authorities) and investigated a variety of abiotic/biotic factors, horse abundance, and socioeconomic variables as predictors for reported equine cases [64]. Avian host species presence outweighed climate, land cover, and demographics. Local heterogeneity was demonstrated spatiotemporally and climatically. Reporting of equine cases followed a northward trend starting in the spring, reaching parity in the middle of the summer, then trending southward in the fall, reflecting the movement of birds and seasonal mosquito activity. Precipitation did correlate to equine cases; however, a nonlinear relationship between activity and the drought index demonstrated that dry conditions actually increased the likelihood of cases, while drought had a negative effect on the likelihood of WNV disease. West Nile virus activity is heterogeneous throughout Central and South America [66], and studies in Brazil found that higher temperatures and lower levels of precipitation were associated with increased seroprevalence in horses [67].

#### 3.1.2. Microscale Factors

Microscale modeling variables are difficult to compare between humans and horses because of vast differences in environmental exposure. Logically, continuous environmental exposure to mosquitoes would overwhelm differences in factors, such as housing and human outdoor activities, socioeconomics, and the occupation of humans and horses. The intersection of risk is best associated with land use. Agriculture and rural locations are usually associated with higher rates of seroprevalence and neuroinvasive disease in humans and horses [60,68,69,70]. In rural environments, horse and human occupation are linked with agriculture, and agricultural workers are more likely to have either higher seroprevalence rates and/or more reported WNV cases. In a Canadian study where rural and urban populations were compared directly, humans residing in rural locations were five times more likely to be seropositive to WNV than their urban counterparts [71]. Crops that attract birds (e.g., orchards, etc.), the presence of forested habitat, and farming that relies on irrigation are commonly and highly associated with WNV seroprevalence and disease in horses as well as humans [72,73]. Where WNV disease risk was studied in urban environments, areas of low population density characterized by higher amounts of vegetation and water features (flow not defined) had higher incidences of reported human cases [48]. The presence of water on a property or nearby is not always associated with disease. In a study of risk factors for WNV neuroinvasive disease in Florida horses, farms with freshwater features consisting of free-flowing water, such as springs and rivers, were negatively associated with the presence of equine disease [74].

### 3.2. Transmission to Humans and Horses

In nature, WNV is transmitted to vertebrate hosts via the bite of a mosquito (Figure 1). Many species of mosquito vectors are competent for transmitting WNV, but Culex sp. are the most competent vectors worldwide [36,75,76,77,78,79]. In general, Cx. pipiens constitutes the majority of WNV-positive mosquito pools in the eastern United States [75], and Cx. tarsalis constitutes the majority in the middle and western United States [76]. In the southeastern United States, Cx. quiquefasciatus and Cx. nigripalpus pools have been found with the highest WNV infection rates [80,81,82]. In Europe, the presence of WNV has two cycles based on the ecosystem, a rural (or sylvatic) cycle usually involving wetland birds and ornithophilic mosquitoes, such as Cx. Molestus, and an urban cycle involving songbirds (Passeriformes) and Cx. pipiens [44,83,84]. Culex univittatus is a main vector of WNV in Africa and the Middle East (with expansion into parts of Europe) [2,85]. Aedes sp. and Anopheles sp. mosquitoes are also considered to be potentially competent vectors and are likely important mammalian feeders [79,86]. The lifecycle of WNV involves primarily avian reservoirs, which can amplify the virus to high titers. High levels of viral amplification occur in many bird species, especially in Passeriformes and Charadriiformes (e.g., shorebirds) [87,88,89,90]. House sparrows and robins are considered important amplifying hosts or “super spreaders” for WNV [88,89,91].

Several species of ticks have been investigated for the potential to transmit WNV; however, the evidence is unclear as to the competency of ticks as vectors. Transstadial transmission was demonstrated in one study of Ixodes ticks but failed to occur in a second study [92]. Infected Argas arboreus ticks experimentally transmitted WNV to chickens [93]. Carios capensis transmitted WNV under experimental conditions to ducklings and several Ornithodoros spp. transmitted WNV under experimental conditions to mice [93,94].

Beyond mosquito transmission, documented infections in humans have been traced back to blood transfusions (reviewed recently by Gimenez-Richarte in 2022) [95,96]. To the authors’ knowledge, there are no reports of transmission of WNV by blood or plasma transfusion in horses. Nonetheless, this remains a possibility since iatrogenic transmission of Theileria equi [97], equine pegivirus [98], and equine infectious anemia virus [99] has been documented. Other routes of transmission documented in humans are through organ transplantation and vertically through breast milk and across the placenta [95,96]. Vertical transmission of WNV has also been reported in horses [100]. Occupational transmission is possible and is typically due to injuries sustained by sharp objects in the laboratory and during postmortem handling of tissues [101,102].

## 4. Comparative Clinical Disease

### 4.1. Viral Kinetics

Human and equine viral kinetics are similar, although more is known about virus dynamics in the horse due to experimental infections. Titers in donated human blood can range from 0.06 to 0.6 PFU/mL [95,103,104,105]. In humans, the estimated period from infection to clinical disease is likely 3 to 14 days, and in one study, the development of symptoms after the onset of viremia averaged 6.9 days [103,105]. Much of the available data regarding WNV infection in horses has been generated by experimental infection, and the course of experimental infection is similar whether horses are challenged intradermally, subcutaneously, intrathecally, or by mosquito feeding [106,107,108,109,110,111,112,113]. One of the first descriptions of experimental infection of horses was published in 1963 [106]. In this early study, six donkeys and three horses received peripheral injections of WNV L2 derived from previous murine brain inoculation. Two of six infected donkeys became viremic with 10^1.5^ LD50 virus/mL (neonatal intracranial inoculation model) for one day on days 4 and 6 post-inoculation; conversely, none of the infected horses became viremic. Infections with L1a, L1b, and more recently, isolated L2 strains have similar kinetics upon peripheral inoculation in horses [107,108,110,111,112,113,114]. In general, horses develop viremia between days 1 and 6 post-inoculation, with titers ranging from 10^1.0^ to 10^3.0^ PFU/mL, usually lasting for a short duration of less than 3 days [107,108,110,111,114]. In work performed with the WNV L1 strain, horses were fed upon by Aedes albopictus mosquitoes infected with 10^6.6^ to 10^7.4^ PFU/mL of virus [107]. These infected horses failed to transmit the virus when fed upon by uninfected mosquitoes between days 3 and 5 post-infection, conclusively demonstrating that the horse is a dead-end host in virulent L1a infections [111]. Viral load in the central nervous system (CNS) of experimentally infected horses is much higher than that of peripheral blood and tissues, measuring between 10^4^ and 10^6.8^ PFU/mg of tissue [110,111,114].

Based on viral culture, WNV is not detectable in the blood of horses and humans once clinical signs become apparent. The paradigm of short duration, low titer viremia prior to the onset of clinical signs is shifting due to enhanced sensitivity of molecular techniques as well as persistent testing in humans and horses [103,105,115]. In one blood donor study, WNV was detectable in human blood for an average of 13.2 days using a transcription-mediated amplification [105]. In a clinical study, WNV RNA was detected by RT-qPCR in the blood of 80% of clinical WNV patients during the first five days of clinical symptoms at cycle threshold (Ct) values below 35, levels which may reflect the continued persistence of live virus [115]. In one study, the L2 virus was detected in the serum and white blood cells (WBCs) of one horse after the onset of neurological symptoms [116].

Originally, WNV infection was considered to be confined to the bloodstream with no detectable shedding in body fluids. Detection of WNV in urine in a human patient after the onset of symptoms was first reported in 2005 [117]. In another study, the urine of WNV patients was positive at Ct values of less than 35 for up to 30 days [115]. These latter findings are important—if molecular detection of WNV is possible in a small subset of humans and horses, genetic sequencing of WNV in humans and horses displaying clinical signs allows for molecular typing in real-time during outbreaks. Detection of the virus in multiple body fluids also provides an opportunity to study the evolution of WNV in the outbred host.

### 4.2. Clinical Syndromes

Not all of the WNV syndromes defined in humans are recognized in horses. Human WNV disease includes West Nile fever (WNF), West Nile encephalitis (WNE), or West Nile acute flaccid paralysis (AFP; polioencephalomyelitis) [118]. West Nile neuroinvasive disease (WNND) usually incorporates both encephalitis and paralysis. In the horse, the syndromes of WNE and AFP are not strictly defined as they are in human diseases. The low prevalence of WNF in horses is likely a reflection of limited arbovirus testing in febrile horses; however, WNF likely occurs, as has been documented in peripheral experimental infections of WNV in horses [107,119]. Furthermore, a model of non-lethal equine infection of WNV was developed in which horses challenged with relatively low doses of L1b developed minimal symptoms but nonetheless had neuronal pathology [113].

### 4.3. Demographics and Outcomes

Demographic risk factors for severe neuroinvasive disease have been well-established in human disease and mainly include age, gender, race, and comorbidities, including diabetes, chronic heart and renal disease, and immunosuppression [120,121,122]. While horses have endocrinopathies and cardiac or renal disease, little is published regarding relevant comorbidities and WNV disease severity in horses. Increased age is a commonly cited risk factor in humans [120,123,124]. In one publication, the incidence of neuroinvasive disease was 10 times higher in patients between the ages of 50 and 59 years old and 43 times higher for those greater than 80 years old [123]. In Romania, the case fatality rate increased from less than 4.3% in patients less than 50 years old to greater than 17% in those older than 70 years [124,125]. This finding holds true for L2 infections where both WNF and WNND increase with age [115]. Age as a risk factor for neuroinvasive disease in horses, as well as a risk factor for severe disease and higher mortality, is unclear based on individual studies. In a selected group of publications that included four or more cases where age was provided, the average age of pooled data from 2891 horses with neuroinvasive disease was 8.5 years (ranging from 0.5 to 32 years) [21,24,116,126,127,128,129,130,131,132,133,134,135,136]. Although there was limited data in most publications regarding the age of the horses that died or were euthanized; in five of the aforementioned studies representing 637 horses, the average age of mortality was 10.37 years (current lifespan is 20–30 years) [24,116,127,130,135]. In two other studies, animals less than one year of age were more likely to die than in other age groups. [134,136] In another, all animals less than one year of age died [128].

Case fatality rates in humans with WNND L1a are generally 8–10% [95,115]. There is variation across Europe in case-fatality rates in L2 infections. In a study of 427 patients in Italy, the case-fatality rate was 22% [115], while the case-fatality rate was 8.8% in a cluster of 57 cases in a study from Romania [137]. The overall mortality was 28% in pooled data of 2993 horses from selected studies [9,22,24,116,126,127,128,129,130,131,132,134,135,136,138,139,140]. Mortality rates calculated using this same data were similar in horses infected with either L1a or L2 viruses, averaging 29% and 34%, respectively.

In human studies, men are typically considered more likely to develop WNV disease than women [95,115,141]. In equine studies, sex ratios vary between studies, with some demonstrating bias toward females [126,130] and in others, male horses [22,24,128,134,135,136]. However, the average of pooled data from 14 studies that included 2182 horses was 50.4% and 49.5% in males and females, respectively [24,116,126,128,129,130,132,136,139,140].

### 4.4. West Nile Fever

Blood donor screening studies and seroprevalence data collected early in the course of WNV emergence in the United States indicate that 20–26% of WNV-exposed humans develop WNF [142,143]. As discussed previously, WNF is characterized by an abrupt onset of febrile illness consisting of clinical signs, such as headache, malaise, skin rashes, fever, myalgia, and/or joint pain [144].

### 4.5. West Nile Neuroinvasive Disease

Neuroinvasive disease is estimated to occur in 0.66–1% of symptomatic human patients [33,142,145]. In horses, approximations vary slightly, but it is estimated that between 10% and 20% of exposed horses develop neuroinvasive disease [107,119]. This number is based on the approximate 1:10 symptomatic to asymptomatic ratio seen in WNV NY99 peripheral challenge studies [107,119] and clinical studies [146].

In a large systematic review of clinical L1 human infection, it was observed that the frequency of specific clinical signs and rates of occurrence for each syndrome were inconsistently reported [118]. WNE presents as a spectrum, from mild short-term confusion and disorientation to severe changes in cognition and sensorium [147,148]. Patients can exhibit various levels of mentation with catatonia and coma [144,145]. Postural tremors occur with primary involvement of the upper extremities [144,145]. This Parkinsonian-like feature also includes bradykinesia, stiffness, and postural instability. Seizures can occur but are infrequent. Meningitis without changes in mentation also occurs and is characterized by fever, headache, and stiff neck [147,149]. Acute flaccid paralysis (AFP) is primarily characterized by the acute onset of flaccid paralysis that is usually asymmetrical with a general loss of spinal reflexes without sensory disruption [147,150,151]. The prognosis for complete resolution of paresis secondary to AFP is guarded [118,152], as AFP can be life-threatening with loss of respiratory muscle function [150].

The clinical features of equine neuroinvasive WNV disease are similar to those observed in human infections. Clinical symptoms in the horse are generally categorized as changes in mentation, locomotor disorders, and/or notable cranial nerve aberrations [127,128]. Changes in mentation can range from depression/obtundation to hyperexcitability. Approximately 30% of horses have some degree of decreased mentation or depression [10,22,116,126,128,129,134,135,136,140] to the point of stupor as well as narcolepsy [10,116,127,129]. Hyperexcitability has also been reported in approximately 30% of cases [9,22,24,116,128,129,133,134,135,136,140]. Horses also exhibit compulsive behaviors, such as constant chewing or walking [22,116,127,128,131,135,136]. Another manifestation is hyperesthesia or hypersensitivity to tactile or auditory stimuli [9,22,24,116,128,129,133,134,135,136,139,140]. Parkinsonian-like features are also prominently displayed in equine disease. Fasciculations manifest as coarse or fine tremors of the upper limbs, neck, and face and occur in approximately 80% of diagnosed horses [9,24,54,116,126,128,129,131,134,135,136,138,139,140]. Similar to humans, these are postural and not noted when lying down. Slow walking, reluctance to walk, and rigidity are also reported [127,128]. Seizure activity is infrequently reported, as in humans [116,131].

In horses, WNV also has a predilection for motor neurons of the cranial nerves [9,116,127,128,129,131,133,134,135,136,139,140]. Most common abnormalities include drooping lip, deviated muzzle, tongue weakness, loss of swallowing reflex, head tilt and listing to one side, and slow pupillary light responses to include combinations of cranial nerves III, VII, VIII, IX, X, and XII. This reflects virus localization to the cranial nerves of the mid- and hindbrain. Limited numbers of horses exhibit blindness [128,131,135,136].

Locomotor deficits are typically multifocal, asymmetric, and may include components of ataxia (approximately 70%) [9,21,24,54,116,126,127,128,129,130,131,132,134,135,136,138] and weakness (45%) [21,24,54,116,126,127,128,129,130,131,132,134,135,136,138]. These two clinical signs reflect both brain and spinal cord disease causing interruption of the sensory and motor tracts in the hindbrain and the spinal cord. Similar to AFP in humans, monoparesis and paraparesis involving one or all limbs in the horse are reported in approximately 40% of horses. Paresis can involve any of the fore or the hind limbs without a consistent pattern from horse to horse [9,22,24,116,126,128,131,133,134,135,136,139]. When unilateral, this generally indicates lower motor neuron disease from viral infection of the grey matter of the ventral and lateral horns of the spinal cord. Approximately 20% of horses can display severe intermittent weakness of front and/or hind limbs causing the horse to be unable to stand (recumbent), temporarily or intermittently kneel, or to dog-sit, depending on which limbs are involved. Forelimb weakness without hindlimb weakness is unique to WNV; it has been described in L1a infections and was overrepresented in one report of L1b infections [136]. In a study of L2 infections, hindlimb weakness was overrepresented [133]. Full persistent recumbency occurs in approximately 30% of cases with or without loss of reflexes [24,116,126,128,129,131,132,133,134,135,136,140]. Many horses that are persistently recumbent are often euthanized due to the poor prognosis for recovery [116,128,133,134,135]. In human cases, persons with AFP or full paralysis also face a grave prognosis for recovery and survival [118].

### 4.6. Extra-Neural Symptoms

Extra-neural symptoms in WNV-affected humans may involve the eye, skin, gastrointestinal tract, heart, and kidneys [153,154,155,156,157]. Involvement of the eye in human infection is well documented [153,154]. Manifestations include chorioretinitis, anterior uveitis, vitritis, and retinal hemorrhage [153,154]. Ophthalmic changes have not been documented in the horse but have been documented in birds of prey and waterfowl [158,159,160,161]. Gastrointestinal signs such as nausea, abdominal pain, and diarrhea have been documented in humans [155,156,157]. Horses can exhibit abdominal pain prior to developing clinical signs. Enterocolitis, colonic impaction, and rectal prolapse have also been documented in the horse [156,157].

### 4.7. Long-Term Recovery

Humans can experience a variety of debilitating cognitive and physical effects from neuroinvasive WNV disease [152]. Even with WNF, lasting fatigue, weakness, and cognitive issues are often self-reported [162]. With WNND, the long-term effects can be substantial, with self-reporting by patients indicating that up to 20% experience fatigue, muscle pain, and headaches [152,163,164,165,166]. Cognitive outcomes include attention and memory deficits and sleep disorders [152,163,164,165,166]. Patients with AFP can have long-lasting paresis and loss of strength [152,163,164,165,166]. Horses can exhibit slow recovery to athletic potential following WNV [127]. Horses can exhibit life-threatening injuries sustained during clinical disease [128].

## 5. Comparative Clinicopathological Features

### 5.1. Clinical Pathology

Clinical pathology is an unrewarding diagnostic tool for use in human and equine WNV patients. Humans with WNE or WNP may have increased peripheral WBC counts. Conversely, prolonged lymphocytopenia has also been reported [147,167]. The most common changes in horses consist of a decreased peripheral lymphocyte count [116,128]. Increased neutrophils have also been observed less commonly [116]. Cerebrospinal fluid (CSF) of WNV-infected humans and horses usually yields a mononuclear pleocytosis with increased protein concentrations [138,156,168,169]. However, in some human patients, a neutrophilic pleocytosis has been noted in the CSF when obtained early in the course of the disease process [167]. Cerebrospinal fluid of equine patients is similar to humans, with most infected animals exhibiting mononuclear pleocytosis that is lymphocytic and occasionally neutrophilic [138,169].

### 5.2. Pathology

#### 5.2.1. Gross Pathology

Descriptions of gross and histopathological lesions in humans as well as horses are usually confined to case reports or limited case series (recently reviewed in Byas 2022) [6,116,133,134,140,170,171,172,173,174,175,176]. Gross pathology is limited to post-mortem examinations of horses infected naturally with WNV. When obvious, lesions in the CNS may include mild to moderate meningeal hyperemia, subdural exudates with fibrin tags, and focal areas of hemorrhage visible on cut sections of the brainstem and spinal cord [116,170,171]. When WNV neuroinvasive disease is suspected in the horse, only examination of the CNS may be performed for safety reasons, which limits our knowledge of WNV pathology in other organs [116,128]. Pulmonary congestion and edema were reported along with hemorrhages in the tissues of the heart of one horse during the 2000 WNV outbreak in Israel [170]. Gross pulmonary lesions have also been described in horses during previous outbreaks in Morocco and Italy [170,173].

#### 5.2.2. Microscopic Pathology

The histopathological findings of WNV neuroinvasive disease are similar in humans and horses, consisting of moderate to severe non-suppurative polioencephalomyelitis in the brain and spinal cord [6,174]. Inflammatory lesions tend to be more severe in aged or immunocompromised humans. In humans, the histopathology is nonspecific and includes perivascular lymphocytic infiltrations, microgliosis, microglial nodules of lymphocytes and histiocytes, and neuronal destruction [151,167,176,177,178,179,180,181]. Limited involvement of the cerebrum is common, although if present, it is usually concentrated in gray matter in both humans and horses [127,151,167,172,176,177,178,179,180,181]. In humans, the most common and severe lesions are in the thalamus, basal ganglia, pons, and medulla oblongata (Figure 2) [145]. Horses have similar changes, with inflammatory cells found most commonly in the thalamus, basal ganglia, midbrain, and hindbrain [128,133,134,140,171,172]. When noted, gliosis and glial nodule formation occur with neuronal degeneration and limited necrosis, except in severe cases (Figure 3a). Spinal cord lesions localize to the ventral and lateral horns in horses and the anterior horns in humans. In some horses, lesions may be found only in spinal tissues [128,171].

In addition to viral culture, detection of WNV antigen and nucleic acids in the brain of humans and horses is required for a confirmatory diagnosis of disease [182,183,184,185]. The most common diagnostic methods consist of immunohistochemistry (IHC) and RT-qPCR. West Nile virus is highest in areas with pathology in the thalamus, basal ganglia, midbrain, and hindbrain (pons and medulla), and IHC staining demonstrates virus-laden neurons within and around lesions (Figure 3b). If IHC is performed in areas of the brain where the pathology is mild to moderate, this procedure is less sensitive than RT-qPCR [186,187]. Limited quantities of virus are typically found in the cerebrum of infected humans and horses (Figure 2 and Figure 3c). Examination of tissues for WNV in the spinal cord should include those segments that reflect neuroanatomical localization of clinical symptoms. The virus is commonly found in the anterior horn of humans and the ventral and lateral horns of horses. Molecular techniques are more sensitive but can also result in false negative testing. While WNV can be isolated from tissues, the need for enhanced laboratory biosafety precludes extensive use of viral culture.

Immunophenotyping has been performed on the CNS lesions found in WNV-infected humans and horses. In several human studies, the predominant cell type within severe lesions consisted of CD8+ T cells and lower numbers of CD4+ T cells [151,174,176,177,179,181,187]. The majority of B cells were confined to the perivascular cuffs [151,174,176,177,179,181,187]. Work performed in experimentally and naturally infected horses demonstrated significant numbers of Iba-1+ microglia, CD3+ T cells, and lesser numbers of MAC387+ macrophages [171,172,188]. The phenotype T cells within tissues of WNV-challenged horses were mixed, composed of approximately equal numbers of CD4+ and CD8+ T lymphocytes. B cells were the least abundant cell populations [188]. Expanded networks of prominent astrocyte cell processes were observed when stained with antibody to the glial fibrillary acidic protein (Figure 3d) [188].

## 6. Diagnosis of WNV

### 6.1. Detection of WNV Antibody

#### 6.1.1. Serosurveillance

Upon infection, horses and humans have a short window of minimally detectable circulating virus; thus, serology is the cornerstone of surveillance, seroprevalence studies, and ante-mortem diagnosis of WNV infection. Serosurveillance is usually performed with a combination of screening tests. ELISA testing consists of in-house or commercially available indirect IgG, IgM capture, and competitive inhibition (CI) formats [189,190,191,192]. The indirect immunofluorescent antibody (IFA) test is used to screen for the closely related Usutu virus [189,193]. This test has value in that it is rapid and less expensive than ELISA test kits; however, the IFA has a higher percentage of false positives [189,193]. The hemagglutination inhibition (HI) test is rarely used and has decreased sensitivity compared to both IgG and IgM ELISA testing in horses [194,195]. In a side-by-side comparison of the HI test to ELISA and PRNT, the endpoint titers failed to differentiate JEV from WNV [194]. All closely related flaviviruses cross-react irrespective of testing format. In regions where multiple flaviviruses circulate, endpoint testing using a neutralization test is required for accurate surveillance [196].

Arbovirus response plans usually rely upon the reporting of positive horses based on serology and/or postmortem findings in horses to identify increased WNV activity. However, the results of laboratory testing are often delayed by days or even weeks. In studies where both humans and horses are simultaneously tested, horses generally have a higher seroprevalence [193,197,198,199,200,201,202], and mapping of WNV antibodies in non-vaccinated populations of horses is predictive of past or recent WNV activity [193,203,204]. The overall usefulness of laboratory-confirmed neuroinvasive horse cases for the prediction of WNV cases in humans is equivocal. In one study, the human disease itself was predictive for equine cases during one season [205]. In another, when horses and human cases were identified in spatial clusters, human cases were not preceded by positive horses [206]. In the United States, the annual case count in horses has even less predictive power due to the widespread use of vaccines. The annual number of reported equine WNV cases has been lower in horses than in humans since 2005. After the release of equine WNV vaccines, WNV infections in horses have dropped to under 500–600 reported cases since 2006 (Figure 4). Since 2005, 20,735 human WNND cases have been reported compared to 6,234 equine WNND cases.

#### 6.1.2. Serodiagnosis of Clinical Disease

Detection of IgM antibodies in the blood and CSF is used for the diagnosis of acute WNV infections in humans [189,207]. In human blood donor studies, the median time from the detection of WNV to seroconversion was 3.9 days [105,167]. While most IgM levels drop within weeks, IgM following WNV infection can persist for months or years in humans. [208,209]. The IgM response in the horse is of shorter duration and generally undetectable at the cut-off dilution of 1:400 by 4–6 weeks after exposure [210,211,212]. Both in-house and commercially available equine IgM capture ELISA formats have high sensitivity and specificity (>88–90%) for a diagnosis of WNV in horses [210,213]. The widespread use of currently available equine WNV vaccines confounds the interpretation of serologic diagnostic tests in the symptomatic horse. Two studies showed no detection of IgM antibody at 1:400 when animals were tested on a weekly basis [210,214]. However, IgM was detected in 13/66 horses post-vaccination in another study [195]. Thus, vaccination history is essential for the interpretation of IgM results in the vaccinated horse. Vaccination with currently inactivated products theoretically does not result in the production of IgM in the CNS due to the blood-brain barrier, and the presence of IgM in the CSF likely indicates neuroinvasive infection due to intrathecal production of antibody [167]. Serum IgM cross-reactivity, as well as serum neutralizing antibody, is common in regions with circulating St. Louis encephalitis virus, yellow fever virus, Zika virus (ZIKV), Dengue virus (DENV), Usutu virus, and JEV [194,207]. Usutu virus and tick-borne encephalitis virus are neuroinvasive flaviviruses that confound the interpretation of testing in both horses and humans in Europe [207] and Africa [215]. Japanese encephalitis virus confounds test results in the east and near-east [194]. Neutralization testing with the comparison of endpoint titers for endemic flaviviruses remains the gold standard for the differentiation of closely related flaviviruses [203].

Neutralizing antibodies to WNV are usually present by the second week of clinical signs and continue to rise over the next several weeks. A four-fold change in neutralizing antibody titers in serum samples obtained 2–4 weeks apart is confirmatory for recent exposure in humans and the nonvaccinated horse [195,211,216]. Vaccination of horses induces the formation of neutralizing antibodies to the E protein of the virus. Knowledge of vaccine history and testing of serial serum samples side-by-side for comparison of endpoint antibody titers is recommended for proper interpretation of WNV-neutralizing antibody test results in the horse.

### 6.2. Other Diagnostic Modalities

Initial clinical workup in humans may include electrodiagnostics and diagnostic imaging. Cranial magnetic resonance imaging (MRI) is more reliable than computed tomography (CT) in determining the presence of CNS inflammation, although both have low specificity for the diagnosis of WNV infection [217,218,219,220,221]. MRI and CT are not used as standard diagnostic procedures for the diagnosis of WNV in horses; however, these are used as ancillary diagnostics to rule out other intracranial diseases and limb and spinal disorders in the horse [222]. Similarly, MRI has been used as part of the diagnostic workup for Eastern equine encephalitis in a dog [223]. Electrodiagnostics did enhance our understanding of WNV by demonstrating interruption of the motor pathways consistent with lower motor neuron disease in human AFP [221].

## 7. Vaccines and Therapeutics against WNV

### 7.1. Vaccines

At present, no WNV vaccine is approved for use in humans, but clinical phase trials have been approved for three vaccines. Two of the vaccines are flavivirus chimeras, in which the prM-E proteins are expressed in either the yellow fever 17D vaccine modified-live virus or DENV, the former being one of the earliest to reach the trial stage. The chimera vaccines stimulate antibody development after one dose. The third vaccine construct is a DNA vaccine that expresses the prM-E proteins, and three doses are required for the development of comparable antibody levels [224]. The 17D chimeric vaccine virus replicates without causing neuropathology, and a version of this vaccine was investigated for safety, immunogenicity, and duration of immunity in horses. Although not available, this vaccine was originally marketed as a single-dose injection which provided 94% protection against grave neurological disease. An inactivated version of this vaccine is currently marketed for use in horses and is labeled to “aid in the reduction in disease, encephalitis, and viremia.” Two other equine vaccine formulations are presently marketed, consisting of two inactivated whole virus vaccines and a recombinant canarypox vectored vaccine [108,109,110,111,112]. One killed vaccine is labeled to “aid in the prevention of encephalitis and viremia.” The other killed vaccine is labeled to “aid in the prevention of West Nile virus.” The canarypox vaccine is labeled to “aid in the prevention of disease, viremia, and encephalitis.” The difference in labeling reflects the challenge model used to test the immunogenicity of each vaccine. Early models utilized mosquito or needle challenge, where most unvaccinated horses develop viremia but limited neurological symptoms [108,224]. Later, vaccine candidates were tested using an intrathecal challenge by injection of the atlanto-occipital space [109,110,111]. In this model, 90–100% of unvaccinated horses developed grave signs of encephalitis. The duration of immunity in vaccinated horses beyond one-year post-immunization is unknown. In one field study, not all horses maintained antibodies for up to one year after the initial primary immunization [195]. In this same study, all horses that had been vaccinated for two consecutive years had neutralizing antibodies for a full year [195].

Vaccines in development for horses include a recombinant vaccine that targets the WNV-EIII domain. In one study, this vaccine demonstrated immunogenicity when combined with an equine CD40 ligand (CD40L) [225]. Another promising equine vaccine candidate under development utilizes an avian paramyxovirus 2 (APMV-2)-vectored WNV construct [226]. Experimentally, this vaccine elicited humoral immune responses in horses as well as birds, thus potentially laying the foundation for a multispecies vaccine platform that could be deployable during epizootics.

### 7.2. Therapeutics

Multiple therapeutics are under development against flaviviruses, including monoclonal antibodies (mAb), peptides, and antivirals [227,228]. Most of the antibody therapeutics in development are for the treatment of ZIKV. Regarding WNV, a combination of 10 mAbs (WNV-86) under investigation results in 50% neutralization of WNV in vitro. Tat-beclin-1 is a peptide containing the HIV-1 Tat protein, which induces autophagy [228]. Endogenous Tat has antiviral activity, and investigation using this synthetic peptide demonstrated up to a 50-fold reduction in WNV titers in cells. In vivo, this peptide experimentally decreases WNV in the brains of infected mice [228].

## 8. Conclusions

Humans and horses are susceptible to grave neuroinvasive disease with shared clinical symptoms, WNV virus dynamics, antibody responses, and clinicopathology [7]. Both species undergo cyclic outbreaks of febrile neurological disease that overlap temporospatially. These similarities can be exploited to leverage and improve surveillance for the detection of WNV outbreaks. The performance of syndromic surveillance defined only by the occurrence of CNS disease in horses and mortality events in birds demonstrated an area under the curve (AUC) of 0.87 for the WNV outbreak detection [11,12]. Exploiting the similarities in neuroinvasive symptoms in humans and horses can enhance the specificity of syndromic surveillance strategies [11,13]. Further harmonization of ancillary testing methods across regions and countries will also result in more accurate and timely detection of WNV outbreaks.

## Figures and Tables

**Figure 1 viruses-15-01230-f001:**
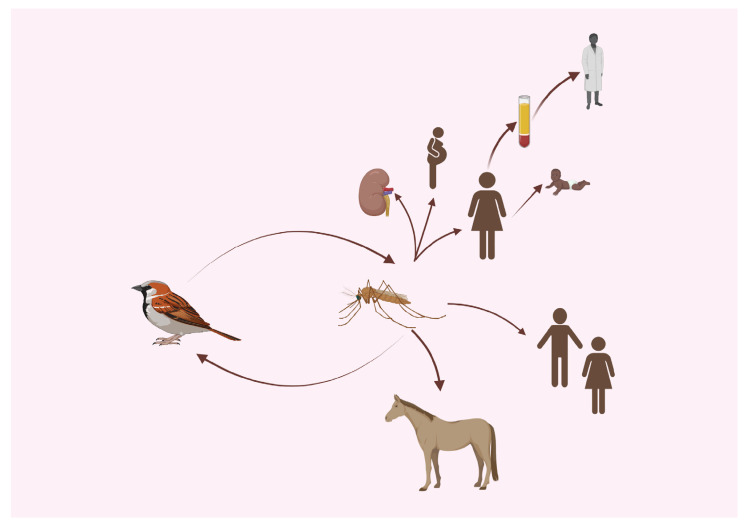
Transmission cycle of West Nile virus to humans and horses. Birds are the main reservoirs of West Nile virus, and upon infection, the virus replicates to levels allowing for transmission to mosquito vectors. In horse and human infections, the viremia is too low for transmission to mosquito vectors; thus, they are dead-end hosts. Other modes of transmission in humans include horizontal transmission by organ transplantation, blood transfusion, and occupationally via sharp instruments. Vertical transmission from mother to child occurs via placenta and breast milk. Created with BioRender.com (accessed on 14 April 2023).

**Figure 2 viruses-15-01230-f002:**
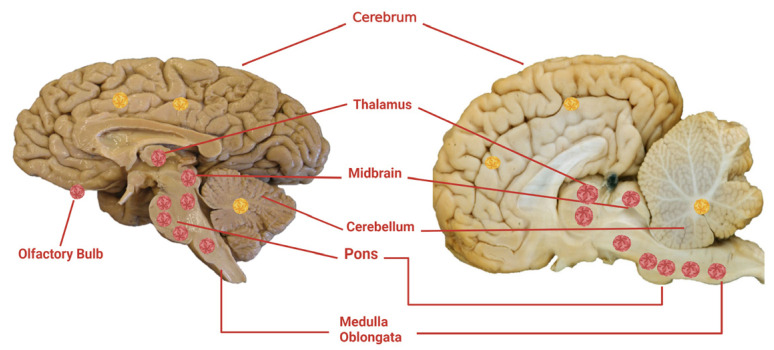
Side-by-side comparison of neuroanatomical localization of West Nile virus in the brain of humans (**left**) and horses (**right**). Red virions represent where the highest levels of virus are typically found, with yellow representing where the lower levels are found (cerebrum and cerebellum). Created with BioRender.com (accessed on 14 April 2023).

**Figure 3 viruses-15-01230-f003:**
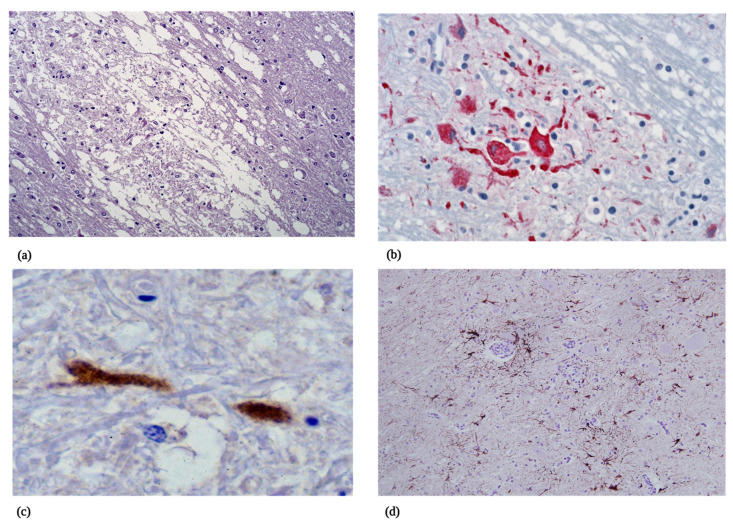
Histopathology of West Nile virus infection in the brain of humans and horses. (**a**) Hematoxylin and eosin-stained section of human brain from a fatal case of WNV demonstrating severe neuronal necrosis and dead neurons, with infiltrating glial cells and leukocytes. (**b**) Immunohistochemistry performed on human central nervous tissue showing WNV antigen (red) in the cytoplasm of neurons and cell processes. (**c**) Immunohistochemistry performed on the brain of a fatal WNV case in a horse showing a single neuron laden with WNV antigen. This was the only stained structure in the cortex of this horse. (**d**) Section of brain from an experimentally infected horse stained with antibody to the glial fibrillary acid protein showing tangles of astrocyte process within glial nodules and associated with perivascular cuffs. (Photos 3a and 3b provided courtesy of Wun-Ju Shieh). Figure created in Biorender.com (accessed on 14 April 2023).

**Figure 4 viruses-15-01230-f004:**
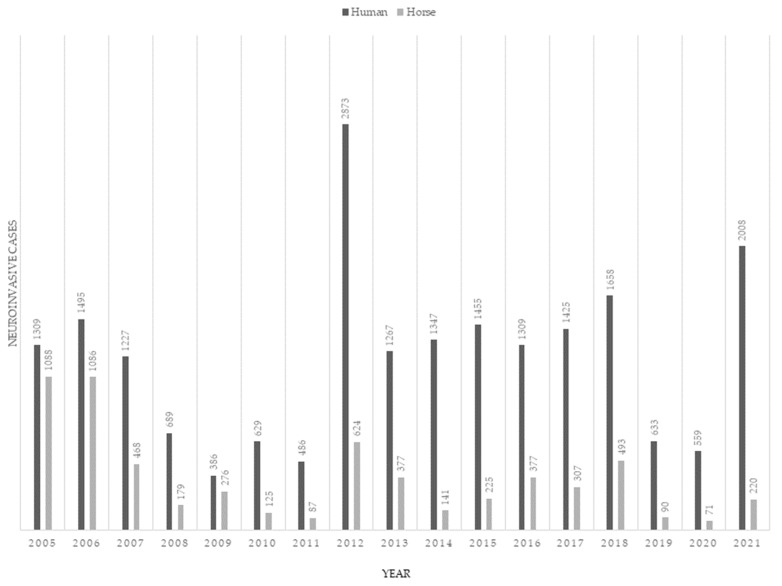
Comparison of United States West Nile virus case counts between humans and horses from 2005 to 2021. (Data obtained from https://www.cdc.gov/westnile/statsmaps/index.html and https://www.aphis.usda.gov/animal_health/downloads/animal_diseases/2021-wnv-report-summary.pdf, accessed on 14 April 2023). Figure created in Biorender.com (accessed on 14 April 2023).

## Data Availability

Data sharing not applicable.

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
