# Peer review of "Comparison of West Nile Virus Disease in Humans and Horses: Exploiting Similarities for Enhancing Syndromic Surveillance"

_viruses, 2023, doi:10.3390/v15061230_

Round 1
Reviewer 1 Report
The review from Schwarz et al. describes the pattern of disease intrahost virus dynamics, evolution of the antibody response, and clinicopathology in human and equine hosts. Similarities were identified and this evidences the need for surveillance for early detection of WNV neuroinvasive disease. The manuscript is well written although minor spell checks should be performed to improve English level.
The manuscript is well written although minor spell checks should be performed to improve English level.
Author Response
We have carefully edited the manuscript. We did not see a report form with specific edits.
Reviewer 2 Report
In this review article, the authors compared the WNV disease in humans and horses. The manuscript is generally well-written.
Please see some comments below.
General comments
Abbreviations should be explained when mentioned for the first time in the manuscript, including the abstract. I suggest using WNV neuroinvasive disease instead of neurological disease. Please correct RT-PCR throughout the manuscript (WNV is an RNA virus).
Abstract
Page 1, line 4: Please correct "the evolution ..."
Line 6: Please correct " ... methods for the early ... "
Introduction
Page 1, line 7: please correct " ... grave neuroinvasive disease ...
Page 1, line 11: Please correct " ... in the prediction ..."
Page 1, line 13: Please correct " ... for the identification ... "
Virus classification and structure
Page 1, line 19: Please correct " ... in the genus Flavivirus ... "
Page 1, line 25: Please correct " pre-membrane ... "
Epidemiology
Page 2, lines 3 and 3: Please correct "neuroinvasive disease" instead of neurological disease.
Page 2, line 4: Please correct " ... late 1990s in ... "
Page 2, line 8: Please correct " ... including the Japanese encephalitis ... "
Page 2, line 20: Please correct " ... upon the scale "
Page 2, lines 33-34: Please correct " ... in the modeling."
Page 2, line 38: Please correct "reservoir hosts, ... "
Page 2, line 41: Please correct " ... evaluated the risk for ... "
Page 3, line 8: Please correct "the intersection of ... "
Page 3, line 19: Please correct " ... had a higher incidence ... "
Page 3, line 20: Please correct "The presence of water ... "
Page 3, line 21: Please correct " ... associated with the disease ... "
Page 3, line 33: Please correct "on the ecosystem: ... "
Page 3, line 41: Please correct " ... considered important ... "
Page 4, line 5: Please correct " ... is possible and is due to ... "
Page 5, line 5: Please correct " ... transmit the virus ... "
Page 5, line 12: Please correct "in the bloodstream prior to the onset ... "
Page 5, line 14: Please correct " ... study, the virus was ... "
Page 5, line 16: Please correct RT-PCR.
Page 5, lines 17-18: Please correct "... reflect the continued persistence of the live virus."
Page 5, line 20: Please correct " ... after the onset ... "
Page 5, line 30: Please correct: " ... are in human diseases."
Page 5, line 32: Please correct " ... likely occurs as ... "
Page 5, line 45: Please correct "50-59-year-olds and ... "
Page 6, line 2: Please correct " ... regarding the age of ... "
Page 6, line 22: Please correct " ... that 20-26% of WNV-exposed ... "
Page 6, line 33: Please correct " ... presents as a spectrum from ... "
Page 7, line 4: Please correct "Fasciculations manifest as coarse ... "
Page 7, line 5: Please correct " ... and face occur in ... "
Page 7, line 7: Please correct " ... to walk have been ... "
Page 7, line 25: Please correct " ... localization of the virus ... "
Page 7, lines 32-22: Please correct " ... loss of reflexes.
Comparative clinicopathological features
Page 8, line 15: Please correct " ... of the disease."
Page 8, line 20: Please correct " ... in humans as well ...
Page 8, line 26: Please correct " ... and edema were reported ... "
Page 8, line 30: Please correct " ... focus on the examination ... "
Page 8, line 45: Please correct " ... with infiltration of ... "
Page 8, line 46: Please correct " ... have a similar"
Page 8, line 48: Please correct" ... surround vessels."
Page 9, line 8: Please correct "In addition to the culture of ... "
Page 9, line 9: Please correct "gold standard"
Page 9, line 13: Please correct " ... in areas with pathology in ... "
Page 9, line 19: Please correct " ... and the ventral and ... "
Page 10, line 23: Please correct " ... antibody to the glial ... "
Diagnosis of WNV
Page 11, line 11: Please correct " ... upon the detection or ... "
Page 11, line 15: Please correct " ... of WNV antibodies ... "
Page 11, line 25: Please correct " ... remained at about ... "
Page 12, line 30: Please correct " ... in the blood ... "
Page 12, line 32: Please correct " ... from the detection of the virus to ... "
Page 12, lines 6 and 12: Please correct "the interpretation ... "
Page 12, lines 12-13: Please correct "as well as ... "
Page 12, lines 15-16: Please correct " ... for the differentiation ... " "Neutralizing antibodies are usually ... "
Page 12, line 20: Please correct "Detection of antibodies in ... "
Page 12, line 22: Please correct "blood-brain"
Page 12, line 23: Please correct "production of IgM ..."
Page 12, line 24: Please correct " ... tick-borne
Page 12, line 27: Please correct " ... induce the formation ... "
Page 12, line 28: Please correct " ... confounds the interpretation of ... "
Page 12, line 37: Please correct " ... used for the diagnosis of ... "
Page 12, line 46: Please correct " ... reach the trial stage."
Page 12, line 48: Please correct " ... for the development"
Page 13, line 11: Please correct " ... with an equine ... "
Page 13, line 14: Please correct " ... as well as birds, ... "
Page 13, line 23: Please correct " ... up to a 50-fold ... "
Conclusions
Page 13, line 26: Please correct " ... to a grave ... "
Page 13, line 27: please correct " ... the evolution of ... "
Page 13, line 30: Please correct " ... for the detection ... "
Page 13, line 35: Please correct " ... defined only by ... "
References
Please check and correct according to the propositions of the journal.
There are some grammatical errors that should be corrected.
Author Response
Please find the attach responses. We have edited this manuscript extensively for style. We also responded to the Reviewer 1 change. We are most grateful for the time this reviewer invested to improve the quality of our writing.
